# DreamVTON: Customizing 3D Virtual Try-on with Personalized Diffusion Models

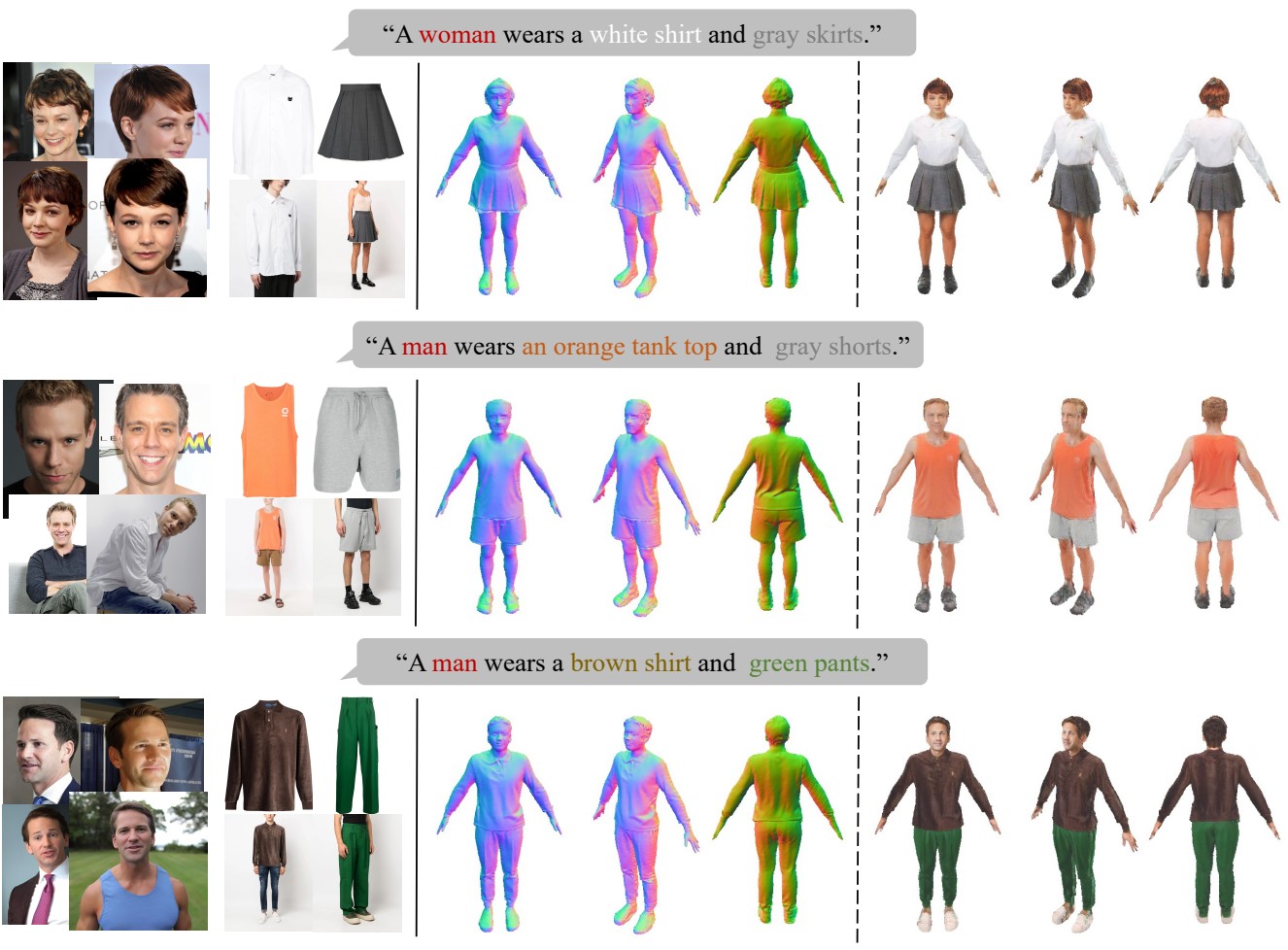

"A woman wears a white shirt and gray skirts."

"A man wears an orange tank top and gray shorts."

"A man wears a brown shirt and green pants."

| Person Images | Clothes Images | 3D Try-on Results (Geometry / Texture) |

**Figure 1: Given a set of person images, clothes images, and a text prompt, our proposed DreamVTON can generate high-quality 3D Humans, wearing customized clothes, keeping the identity and clothes style.**

## ABSTRACT

Image-based 3D Virtual Try-ON aims to sculpt the 3D human according to person and clothes images, which is data-efficient (i.e., getting rid of expensive 3D data) but challenging. Recent text-to-3D methods achieve remarkable improvement in high-fidelity 3D human generation, demonstrating its potential for 3D virtual try-on. Inspired by the impressive success of personalized diffusion models (e.g., Dreambooth and LoRA) for 2D VTON, it is straightforward to achieve 3D VTON by integrating the personalization technique into the diffusion-based text-to-3D framework. However, employing the personalized module in a pre-trained diffusion model (e.g., StableDiffusion (SD)) would degrade the model's capability for multi-view or multi-domain synthesis, which is detrimental to the geometry and texture optimization guided by Score Distillation Sampling (SDS) loss. In this work, we propose a novel customizing 3D human try-on model, named **DreamVTON**, to separately optimize the geometry and texture of the 3D human. Specifically, a personalized SD with multi-concept LoRA is proposed to provide the generative prior about the specific person and clothes, while a Densepose-guided ControlNet is exploited to guarantee consistent prior about

body pose across various camera views. Besides, to avoid the inconsistent multi-view priors from the personalized SD dominating the optimization, DreamVTON introduces a template-based optimization mechanism, which employs mask templates for geometry shape learning and normal/RGB templates for geometry/texture details learning. Furthermore, for the geometry optimization phase, DreamVTON integrates a normal-style LoRA into personalized SD to enhance normal map generative prior, facilitating smooth geometry modeling. Extensive experiments show that DreamVTON can generate high-quality 3D Humans with the input person, clothes images, and text prompt, outperforming existing methods.

## CCS CONCEPTS

• **Computing methodologies → Computer vision tasks**.

## KEYWORDS

3D Virtual Try-on, 3D Human, Personalized Diffusion Models

## 1 INTRODUCTION

The task of Virtual Try-ON (VTON), to transfer a clothing item onto a specific person, has been explored a lot in recent years due to its promising potential to revolutionize the industry of e-commerce and fashion design. The image-based 2D solutions take person and clothes images as inputs and achieve virtual try-on via 2D generative models [13, 22, 48, 54]. Although the advanced 2D solutions [2, 11, 14, 20, 33, 57, 67] can synthesize compelling results within particular viewpoint (e.g., front view), they fail to display try-on results for arbitrary observed viewpoint, which is commonly required in the real-world scenarios. Traditional 3D solutions model the try-on results in the 3D space, thus providing a more comprehensive and attractive perception of clothes fitting. However, most of these solutions [3, 15, 17, 32, 42] rely upon the 3D scanning equipment or labor-intensive manual annotation, making them much resource-hungry compared with the image-based 2D counterparts. The pros and cons of the existing 2D and 3D solutions inspire us to rethink whether sculpting the 3D try-on human by simply using the person and clothes images is possible.

Recently, the extraordinary success of diffusion models [46, 48, 50] for text-to-image (T2I) has largely prompted the development of high-quality 3D content generation [8, 36, 43, 45, 52, 56], whose optimization of the 3D representation is guided by 2D generative priors from the pre-trained T2I diffusion model (e.g., StableDiffusion(SD) [48]) by using Score Distillation Sampling (SDS) loss [43]. Previous 3D human generation works explore this diffusion-based 3D generation framework to sculpt a 3D human according to textual descriptions [7, 25, 26, 28, 30, 35, 62] or reference human images [27, 61]. Despite the significant advancement in high-quality 3D modeling, these methods can not be directly adapted to 3D virtual try-on, because they neither take the clothing items as input nor consider the clothing manipulation during the 3D modeling procedure. Some diffusion-based methods have explored the potential of employing lightweight personalized modules (i.e., LoRA [24]) in SD for 2D virtual try-on. As shown in Figure 2 (a), by using several clothes images for LoRA fine-tuning, the personalized SD can generate photo-realistic fashion models wearing specific clothes.

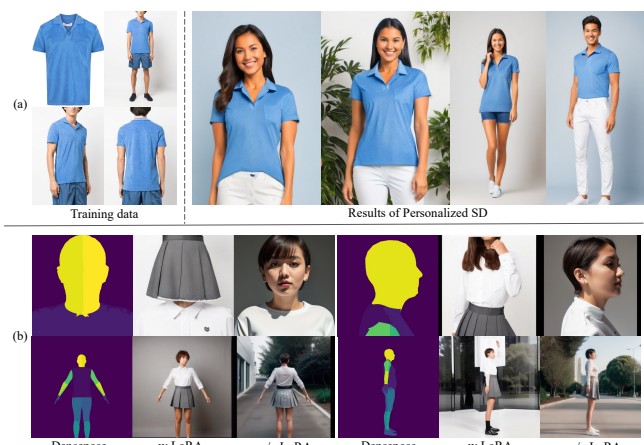

**Figure 2: (a) Try-on results of the personalized SD. (b) Visual comparison between results of SD with and without LoRA directly. Using LoRA directly will reduce the capability of SD for multi-view synthesis.**

Considering the benefits of 2D and 3D generation, it is straightforward to achieve 3D virtual try-on by integrating the personalized SD with the diffusion-based 3D generation framework.

However, introducing LoRA into SD would degrade the model's capability for multi-view generation, since it is trained using rare images within limited viewpoints. As shown in Figure 2 (b), for some observed viewpoints, given the same prompt, integrating SD with LoRA would generate wrong results or directly crash, while SD without LoRA can generate realistic results conforming to the input densepose [16]. The degraded ability for multi-view synthesis results in inconsistent generative priors across various viewpoints and further influences the 3D optimization procedure, leading to coarse 3D geometry and blurred texture. Therefore, it is non-trivial to integrate the personalized SD into a diffusion-based 3D generation framework for image-based 3D virtual try-on.

To target the challenges, we propose a novel diffusion-based 3D human generation framework, named DreamVTON, to sculpt the 3D human by simply taking several person images, clothes images, and a text prompt as inputs (see Figure 1). Specifically, our DreamVTON inherits the advanced two-stage 3D generation framework [8, 25, 27], the first stage optimizes the DMTet-based [10, 51] 3D representation, while second stage optimizes the texture. During the geometry and texture optimization procedures, DreamVTON introduces a multi-concept LoRA to provide generative priors about the specific person and clothes. Besides, inspired by AvatarVerse [62], DreamVTON employs a Densepose-guided ControlNet [63] to provide consistent priors about body pose across various viewpoints. To avoid inconsistent generative priors from personalized SD dominating the 3D optimization procedure, DreamVTON proposes a template-based optimization mechanism, which employs mask templates for precise geometry shape learning and normal/RGB templates for precise geometry/texture details learning. The personalized SD generates the RGB templates within several pre-defined viewpoints, while the mask and normal templates are derived from the RGB templates by using the off-the-shelf mask

and normal predictors. Moreover, to enhance the 3D geometry perception of the personalized SD during the geometry optimization procedure, DreamVTON introduces a normal-style LoRA which can facilitate the personalized SD to provide more powerful prior about the normal map, leading to smoother geometry modeling.

Overall, the main contributions can be summarized:

- We propose a diffusion-based 3D virtual try-on framework, named DreamVTON, which employs personalized SD with multi-concept and normal-style LoRAs to provide powerful generative priors for 3D human optimization.
- We jointly exploit the SDS loss and template-based optimization mechanism for high-quality 3D human modeling.
- We further introduce a normal-style LoRA into personalized SD for smoother geometry.
- Extensive experiments show that DreamVTON can generate high-quality 3D try-on results, consistent with the input images, and outperform other 3D human generation methods.

## 2 RELATED WORKS

### 2.1 2D/3D Virtual Try-on

Most Virtual Try-ON (VTON) methods [2, 14, 19, 20, 33, 55, 57, 59] explore 2D VTON and aim to transfer an in-shop garment onto a specific person. Generally, they employ a two-stage framework to process garment deformation and try-on generation separately, in which the former uses the Thin Plate Splines [5] (TPS) or flow-based [65] network to model geometry deformation, while the latter employs generative models like Generative Adversarial Network [13] or Diffusion Model [48] to synthesize the try-on results. TryonDiffusion [67] introduces an implicit warping mechanism and processes clothes warping and try-on generation within a single diffusion network. Traditional 3D VTON methods [6, 15, 17, 32, 42] relies on the 3D scan equipment or cloth simulation to generate geometric representations of high precision. Learning-based methods [4, 39, 40, 66] employ differentiable rendering to dress the SMPL [37] model with desired garment mesh. M3D-VTON [64] proposes a depth-based 3D VTON framework to lift the 2D VTON results to 3D. Differently, we handle image-based 3D VTON by using the powerful generative model, which can integrate the complementary advantages of 2D and 3D VTON.

### 2.2 Diffusion-based 3D Human Generation

Diffusion-based 3D human generation methods[7, 28, 61] aim to generate 3D humans, using text prompts or reference images as input. They apply SDS-based optimization[43] to progressively generate 3D humans from initial shape often parameterized by SMPL[37]. TADA[35] and TeCH[27] deploys SMPL-X[41] expressing 3D human with more detail. Pose-aware neural human representation imGHUM[1] used to generate the human body in DreamHuman[30]. AvatarBooth[61] employs DreamBooth[49] to inject specific identity information into SD, enhancing identity consistency in the personalized 3D human body generation process. Both DreamWaltz[26] and AvatarVerse[62] leverage Pose ControlNet[63] to obtain detailed human body models. HumanNorm [25] introduces a normal-aligned diffusion model that allows for custom identities and poses using normal maps in specific regions. Our DreamVTON is a pose-aware 3D VTON pipeline that keeps face identity and clothes style.

## 2.3 Personalized Diffusion Model

Dreambooth[49] proposes fine-tuning the network using a small set of subject-specific images, which learns specific objectives of the object with a unique identifier. Textual inversion[9] achieves efficient personalization by optimizing text embeddings, which is used to guide the creation of personalized images during inference. SVDiff[18] introduces an innovative approach by optimizing the singular values of weight matrices within the model. Custom Diffusion[31] focuses on fine-tuning the key and value projection matrices of cross-attention layers, and can jointly train for multiple concepts or combine multiple fine-tuned models through closed-form constrained optimization. LoRA[23] introduces novel styles or concepts into pre-trained text-to-image models by optimizing low-rank approximations of weight residuals. However, DreamVTON addresses the 3D personalized challenge, enabling the generation of diverse clothes while preserving identity.

## 3 METHODOLOGY

The image-based 3D Virtual Try-ON aims to sculpt the 3D digital human using several images of a specific person and clothes items. To achieve this, we propose DreamVTON, a personalized 3D human generation framework (Sec. 3.1) that collaboratively employs multi-concept LoRA and Densepose-guided ControlNet to provide the particular generative priors for the 3D optimization procedures of geometry and texture. To avoid the inferior generative priors from personalized modules (i.e., multi-concept LoRA) dominating the optimization procedure, DreamVTON employs a template-based optimization mechanism (Sec. 3.2) to facilitate realistic geometry and texture modeling. Besides, to further enhance the perception of the 3D geometry, DreamVTON introduces a normal-style LoRA (Sec. 3.3) into the personalized SD. An overview of DreamVTON is displayed in Figure 3.

### 3.1 Personalized 3D Human Generation

**Two-stage 3D generation framework.** To efficiently model high-quality 3D try-on digital human, our DreamVTON inherits the advanced two-stage 3D human generation framework [8, 25, 27], in which the 3D geometry and texture are optimized separately by using Score Distillation Sampling (SDS) [43] to distill the generative priors from the pre-trained Stable Diffusion (SD) [48] $\epsilon_\phi$.

For geometry modeling, DreamVTON utilizes a MLP network $\Psi_g$ to parameterize the DMTet-based [10, 51] geometry representation $(V_T, T)$, in which $\Psi_g$ is trained to predict the Signed Distance Function (SDF) value $s_i$ and the deformation offset $\triangle v_i$ for each vertex $v_i \in V_T$ in tetrahedral grid $T$. During training, DreamVTON first employs an initialization phase to fit $T$ onto an A-pose SMPL [37] mesh $M_{smpl}$, by using the following object function:

$$\mathcal{L}_g^{init} = \sum_{p_i \in \mathbf{P}} \left\| s\left(p_i; \Psi_g\right) - SDF\left(p_i\right) \right\|_2^2, \quad (1)$$

where $\mathbf{P}$ is a point set randomly sampled around the surface of $M_{smpl}$, and $SDF\left(p_i\right)$ is the pre-calculated SDF value. Then, DreamVTON employs the SDS-based optimization mechanism to sculpt the geometry details. To be specific, DreamVTON conducts differentiable rendering onto DMTet mesh to obtain a normal map $\mathbf{n}$, which will then be passed to the pre-trained SD to calculate the normal

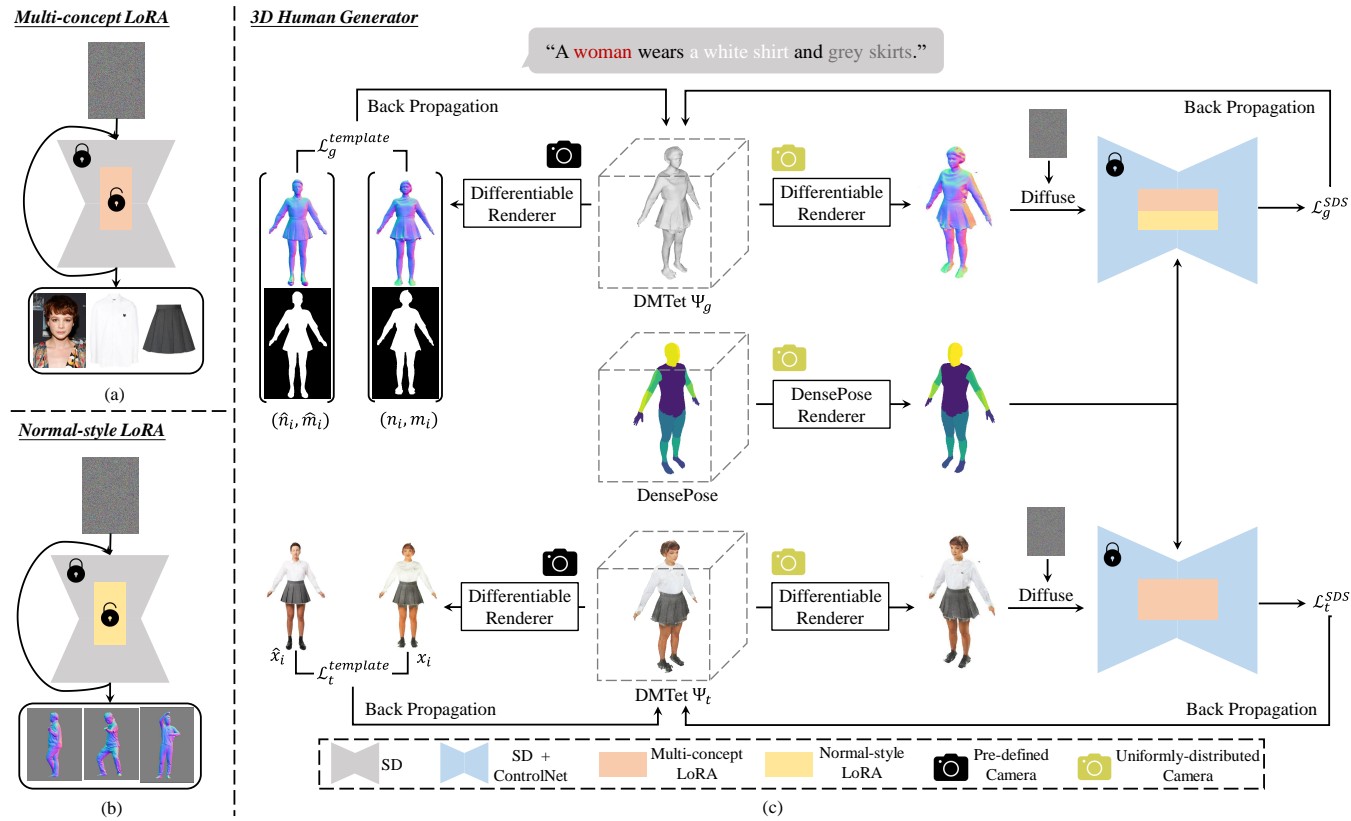

**Figure 3: Overview of our DreamVTON. DreamVTON can generate realistic-looking 3D humans given person images, clothes images, and a text prompt. We disentangle the 3D try-on into geometry and appearance learning and design a Multi-concept LoRA and a Normal-style LoRA. Furthermore, we employ a templated-based optimization to achieve high-quality geometry and detailed texture.**

map SDS loss as follows:

$$\mathcal{L}_g^{\text{SDS}} = \mathbb{E} \left[ w(\mathbf{t}) \left( \epsilon_\phi \left( \mathbf{z}_t^{\mathbf{n}}; \mathbf{c_n}, \mathbf{t} \right) - \epsilon \right) \frac{\partial \mathbf{n}}{\partial \psi_g} \frac{\partial \mathbf{z}_t^{\mathbf{n}}}{\partial \mathbf{n}} \right], \quad (2)$$

where $\mathbf{z}_t^{\mathbf{n}}$ is the latent code of $\mathbf{n}$ with $\mathbf{t}$-step noising, $\mathbf{c_n}$ is the embedding of normal map prompt extracted by CLIP [44], and $\psi_g$ is the parameters of $\Psi_g$.

For texture modeling, DreamVTON utilizes another MLP network $\Psi_t$ to parameterize the material model and uses the Physically-Based Rendering derived from Fantasia3D [8] to obtain the rendered RGB image $\mathbf{x}$. During training, DreamVTON feeds $\mathbf{x}$ into pre-trained SD to calculate the image SDS loss as follows:

$$\mathcal{L}_t^{\text{SDS}} = \mathbb{E} \left[ w(\mathbf{t}) \left( \epsilon_\phi \left( \mathbf{z}_t^{\mathbf{x}}; \mathbf{c_x}, \mathbf{t} \right) - \epsilon \right) \frac{\partial \mathbf{x}}{\partial \psi_t} \frac{\partial \mathbf{z}_t^{\mathbf{x}}}{\partial \mathbf{x}} \right], \quad (3)$$

where $\mathbf{z}_t^{\mathbf{x}}$ is the latent code of $\mathbf{x}$, $\mathbf{c_x}$ is the embedding of image prompt, and $\psi_t$ is the parameters of $\Psi_t$.

**Personalized SD for image-based 3D VTON.** Although existing methods [25, 27] based on the above two-stage framework can obtain high-quality 3D digital human, they can not be adapted to image-based 3D VTON, because they are incapable of handling clothes inputs. To address this problem, DreamVTON introduces a multi-concept LoRA to inject the knowledge of the specific person

and clothes into pre-trained SD, which will provide the generative priors of clothes and person for 3D optimization. The multi-concept LoRA is trained by jointly using person images and clothes images. Person images and clothes images separately provide the identity and clothes information for virtual try-on. As shown in Figure 4 (a), the person images contain several person images of the same person, while the clothes image contains in-shop clothes and fashion models wearing the particular clothes. As for text prompts, DreamVTON employs the visual-language model BLIP [34] to generate captions for each training image. During the inference stage, the text prompt is constructed by extracting the principal concept of each image set, such as *"a woman wears a white shirt and grey skirt."* We display additional examples of the text prompts used for training and inference in the supplementary material. Besides, inspired by AvatarVerse [62], DreamVTON exploits a Densepose-guided ControlNet [63] to provide consistent generative priors about body pose across various viewpoints. Therefore, by employing multi-concept LoRA and Densepose-guided ControlNet, the pre-trained SD item in Eq. 2 and Eq. 3 should be modified to $\tilde{\epsilon}_\phi \left( \mathbf{z}_t^{\mathbf{n}}; \mathbf{c_n}, \mathbf{t}, \mathbf{p} \right)$, where $\tilde{\epsilon}_\phi$ refers to SD with LoRA and ControlNet, while $\mathbf{p}$ refers to DensePose rendered from a SMPL mesh within current camera pose and translation.

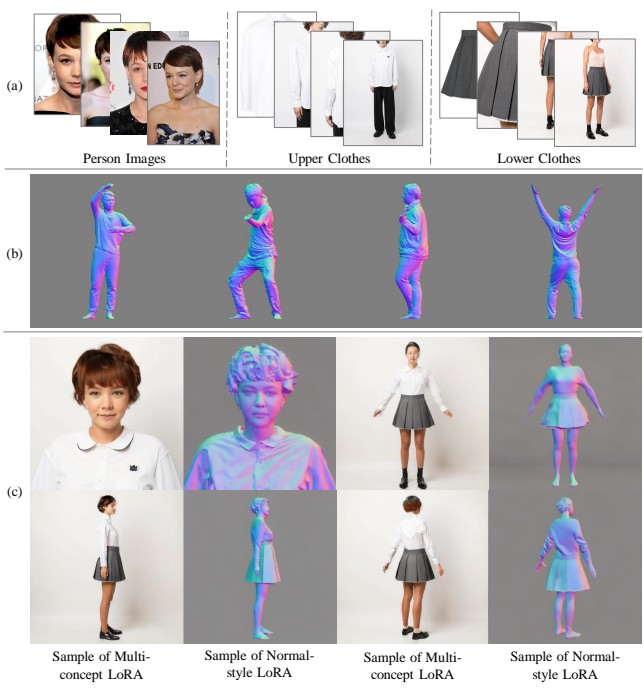

Figure 4: (a) Sample of training data for multi-concept LoRA. (b) Sample of training data for normal-style LoRA. (c) Sample results of multi-concept LoRA and Normal-style LoRA.

## 3.2 Template-based 3D Optimization Mechanism

SDS-based 3D optimization mechanism commonly samples camera poses uniformly distributed in the 3D space, which enables the generative model (e.g., SD) to provide generative priors from as many viewpoints as possible, and thus facilitates comprehensive 3D optimization. However, when employing the multi-concept LoRA into SD, the capability of multi-view synthesis largely degrades (as shown in Figure 2), since LoRA is trained by using rare images within limited viewpoints. The degradation of personalized SD in multi-view synthesis results in inconsistent generative priors across various viewpoints, which is detrimental to 3D optimization. Therefore, simply using the SDS loss from personalized SD would not be an optimal optimization method for image-based 3D VTON.

To prevent inconsistent generative priors across multiple views from dominating the optimization of 3D models, our DreamVTON introduces a template-based optimization mechanism to facilitate precise geometry and texture learning. Specifically, DreamVTON first employs personalized SD (i.e., with multi-concept LoRA and Densepose-guided ControlNet) to generate RGB results within $N$ pre-defined viewpoints, which can synthesize realistic results. The generated results are regarded as the RGB templates $\{\hat{\mathbf{x}}_\mathbf{i}\}_{i=1}^N$, which will then be passed into the off-the-shelf parsing predictor [12] and normal map predictor [58] to obtain the mask templates $\{\hat{\mathbf{m}}_\mathbf{i}\}_{i=1}^N$ and normal templates $\{\hat{\mathbf{n}}_\mathbf{i}\}_{i=1}^K$.

During geometry learning, to guarantee DMTet is optimized into the correct geometry shape, DreamVTON calculate Mean Square Error (MSE) loss $\mathcal{L}_g^m$ between the template masks $\{\hat{\mathbf{m}}_\mathbf{i}\}_{i=1}^N$ and their corresponding rendered masks $\{\mathbf{m}_\mathbf{i}\}_{i=1}^N$ (rendered under the same camera poses with those of templates), which can be formulated as follows:

$$\mathcal{L}_g^m = \sum_{i=1}^N \|\mathbf{m}_\mathbf{i} - \hat{\mathbf{m}}_\mathbf{i}\|_2^2, \tag{4}$$

Besides, to facilitate learning of geometry detail, DreamVTON introduces reconstruction losses between $\{\hat{\mathbf{n}}_\mathbf{i}\}_{i=1}^N$ and their corresponding rendered normal maps $\{\mathbf{n}_\mathbf{i}\}_{i=1}^N$, which consist of a MSE loss $L_g^{mse}$ and a perceptual loss [29] $L_g^{per}$, and can be formulated as follows:

$$\mathcal{L}_g^{mse} = \sum_{i=1}^N \|\mathbf{n}_\mathbf{i} - \hat{\mathbf{n}}_\mathbf{i}\|_2^2, \tag{5}$$

$$\mathcal{L}_g^{per} = \sum_{i=1}^N \sum_{j=1}^5 \lambda_j \left\| \gamma_j(\mathbf{n}_\mathbf{i}) - \gamma_j(\hat{\mathbf{n}}_\mathbf{i}) \right\|_1, \tag{6}$$

where $\gamma_j$ denotas the $j$-th feature map in a pre-trained VGG [53] network. Similarly, during texture learning, to facilitate learning of texture detail, DreamVTON exploits the MSE loss $L_t^{mse}$ and perceptual loss $L_t^{per}$ between $\{\hat{\mathbf{x}}_\mathbf{i}\}_{i=1}^N$ and their corresponding rendered RGB images $\{\mathbf{x}_\mathbf{i}\}_{i=1}^N$.

It is worth noting that, since the normal and RGB templates are derived from the same generated results, the geometry and texture details on each normal-RGB template pair (i.e., templates rendered under the same camera pose) are strictly aligned. By using the detail-aligned templates for geometry and texture learning, DreamVTON is capable of generating geometry-texture consistent 3D human.

By jointly using the SDS-based and template-based optimization mechanisms, the overall object functions for geometry and texture optimization can be formulated as follows:

$$\mathcal{L}_g = \mathcal{L}_g^{\text{SDS}} + \lambda_g^m \mathcal{L}_g^m + \lambda_g^{mse} \mathcal{L}_g^{mse} + \lambda_g^{per} \mathcal{L}_g^{per}, \tag{7}$$

$$\mathcal{L}_t = \mathcal{L}_t^{\text{SDS}} + \lambda_t^{mse} \mathcal{L}_t^{mse} + \lambda_t^{per} \mathcal{L}_t^{per}, \tag{8}$$

where $\lambda_g^*$ and $\lambda_t^*$ are the trade-off hyperparameters.

## 3.3 Normal-style LoRA for Geometry Learning

During the geometry optimization stage, since DreamVTON employs the rendered normal map to calculate the SDS loss, the personalized SD is designed to provide the normal-style generative prior for geometry optimization. However, when introducing the multi-concept LoRA, the personalized SD's capability for normal map synthesis degrades a lot, since LoRA is trained by using seldom RGB images. To address this issue, our DreamVTON introduces normal-style LoRA into personalized SD to enhance the capability for normal map synthesis. Specifically, the normal-style LoRA is trained with 2,000 text-annotated normal maps, in which the normal maps are rendered from the THUman2.0 dataset [60], while the text prompts are extracted by BLIP [34]. Once trained, the normal-style LoRA is integrated into the personalized SD and jointly used for geometry optimization.

As shown in Figure 4 (c), given the same prompt "a woman wears a white shirt and grey skirt, **with normal map style**.", multi-concept LoRA can only generate realistic RGB images, while normal-style can generate results with normal map style, demonstrating

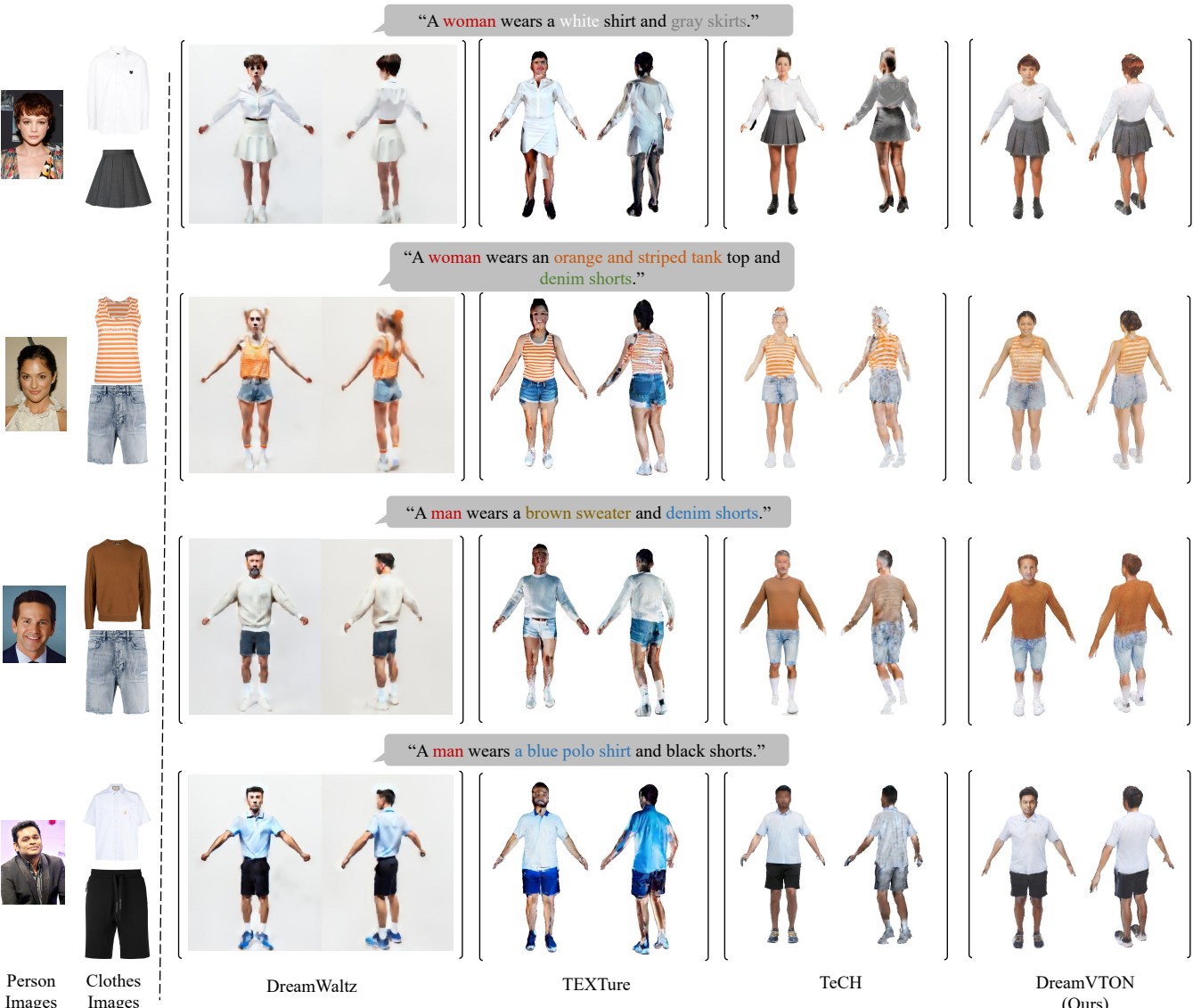

**Figure 5: Qualitative Comparisons. Using the same clothes images, person images, and text prompt as inputs, our method achieves superior results.**

that introducing the normal-style LoRA, could improve the the personalized SD's normal synthesis capability.

## 4 EXPERIMENTS

We first introduce the implementation details of DreamVTON (Sec. 4.1), which contains the dataset description, training configuration, camera sampling strategy, and template selection mechanism. Then, we compare DreamVTON with existing 3D human generation methods qualitatively and quantitatively (Sec. 4.2, Sec. 4.3, and Sec. 4.4). Finally, we conduct ablation studies to verify the effectiveness of the proposed modules in DreamVTON (Sec. 4.5).

### 4.1 Implementation Details

**Dataset Description.** Since there is no existing dataset tailored for the task of image-based 3D virtual try-on, we collect a new dataset from the internet, which comprises images of 10 various individuals and 33 clothes items (i.e., upper clothes, lower clothes, dresses.). Specifically, most of the individual images are portrait images and each individual contains about 10 portrait images. On the other hand, the clothes images contain in-shop clothes and fashion models wearing particular clothes. By matching the portrait and clothes images, we can obtain the person-clothes pairs used for the training of image-based 3D virtual try-on. In our experiments, we construct 18 person-clothes pairs, which is then used in our 3D try-on experiments. Some visual examples can be found in Figure 4.

**Training configuration.** The geometry network $\Psi_g$ and texture network $\Psi_t$ are trained 15000 and 3000 iterations, respectively. The training procedure of $\Psi_g$ can be further divided into 2000 iterations SDF initialization phase and 13000 iterations DMTet optimization phase. During training, the batch size on each GPU is set to 1 and both networks are trained by using AdamW [38] optimizer. The learning rate for $\Psi_g$ and $\Psi_t$ are set to 1e-3 and 1e-2, respectively. Both $\Psi_g$ and $\Psi_t$ are trained on 1 NVIDIA 4090 GPUs.

**Camera sampling strategy.** For geometry learning, at the beginning of DMTet optimization, the randomly sampled cameras are located in a position that can cover the full human body. To sculpt the geometry details, after 1200 iterations, the cameras are posed to positions that focus on various local regions (i.e., head, upper body, and lower body), within which SD can provide more detailed generative prior for geometry optimization. For texture learning, the local cameras are employed at the beginning to enhance the learning of texture details.

**Selection of geometry/texture templates.** For geometry learning, we employ eight mask templates $\{\hat{\mathbf{m}}_i\}_{i=1}^{8}$ and two normal templates $\{\hat{\mathbf{n}}_i\}_{i=1}^{2}$ for optimization, in which $\{\hat{\mathbf{m}}_i\}_{i=1}^{8}$ are sampled uniformly around the human body while $\{\hat{\mathbf{n}}_i\}_{i=1}^{2}$ is composed of the front and back view normal maps. For texture learning, we utilize three RGB templates $\{\hat{\mathbf{x}}_i\}_{i=1}^{3}$, which contain front and back views of full body images and one head image. The head image is used to enhance the texture detail around the face region.

## 4.2 Qualitative Results

We compare our DreamVTON with three existing 3D human generation methods, namely DreamWaltz [26], TEXTure [47], and TeCH [27]. Since DreamWaltz and TEXTure take as input merely the text prompt, we use the constructed text prompts (used by our DreamVTON) for them. For TeCH, since it receives the text and image inputs, except for the constructed text prompt, we also feed the front-view RGB template (used by our DreamVTON) for it. Figure 5 displays the qualitative comparison of DreamVTON with the baselines. By simply receiving the text prompt as model input, DreamWaltz, and TEXTeure can generate 3D humans with similar clothes types to the input clothes images. However, they fail to preserve the texture detail or clothes color. For example, for the white-tshirt-grey-skirts case in the first case, they fail to generate grey skirts. TeCH [27] takes both text and images as inputs, thus is capable of preserving the clothes details in front view. However, it fails to generate realistic texture in the back view and face region, since it ignores the guidance about the back view and face. In comparison, DreamVTON can not only preserve the clothes information in arbitrary view but also generate a realistic face, demonstrating outperforms the compared methods.

## 4.3 Quantitative Comparison

Inspired by HumanNorm [25], we choose CLIP-similarity [44] and FID [21] to evaluate the generated quality of the 3D try-on results, in which FID measures the realism of rendered results while CLIP-Similarity measures the similarity between the rendered results from arbitrary viewpoint and the particular reference images. Specifically, for each test case, we first employ the personalized SD (Densepose-based ControlNet + Multi-concept LoRA) to generate 8

**Table 1: Quantitative Comparisons in the collected datasets.**

|  | DreamWaltz | TEXTure | TeCH | DreamVTON |
|---|---|---|---|---|
| FID ($\downarrow$) | 171.6 | 163.4 | 142.7 | **140.8** |
| CLIP-similarity ($\uparrow$) | 0.596 | 0.613 | 0.655 | **0.665** |

**Table 2: User study results about the 3D generation quality in terms of Geometry, Texture and ID Fidelity.**

| Preference($\uparrow$) | DreamWaltz | TEXTure | TeCH | DreamVTON |
|---|---|---|---|---|
| Geometry | 0.5% | 18.0% | 2.6% | **78.8%** |
| Texture | 1.3% | 1.3% | 6.0% | **91.5%** |
| ID Fidelity | 1.8% | 1.4% | 2.9% | **94.0%** |

2D try-on results, of which the camera viewpoints are uniformly distributed around the human body. Then, we employ similar but much denser cameras to render another 100 images from the learned 3D try-on results. we obtain 100 rendered images from the learned 3D try-on results. During Calculating FID, we regard the 2D generated images as the ground truth and measure the distribution similarity between the pseudo ground truth (i.e., 2D generated results) and the rendered results. During calculating CLIP-similarity, we regard the SD-generated images as reference images and calculate the average CLIP distance between the rendered images and reference images. As reported in Table 1, DreamVTON obtains the lowest FID score, which indicates the images rendered by DreamVTON are most closely aligned to the generated results of the personalized SD. Besides, DreamVTON obtains the highest CLIP-similarity, which further demonstrates the rendered images are more consistent with the reference images. Both of the reported scores illustrate the superiority of DreamVTON over existing baselines.

## 4.4 User Study

We evaluate our proposed DreamVTON's performance against other methods using the user study. As reported in Table 2, a higher score for user evaluation indicates that humans preferred the performance, our proposed DreamVTON outperforms all the compared methods. In particular, our proposed DreamVTON significantly outperforms the compared methods in terms of texture realism and ID fidelity, with 91.5% and 94% of users, preferring to choose our model. Regarding the quality of the geometry, 78.8% of users still prefer to choose our method. Overall, Table 2 demonstrates the effectiveness of our method, which outperforms the other methods on texture, ID fidelity, and geometry, respectively. This means that our proposed DreamVTON can generate more realistic-looking 3D humans that are preferred by users, wearing different clothes with accurate 3D geometry and detailed textures.

## 4.5 Ablation Study

We conduct ablation studies to validate the effectiveness of template-based optimization mechanisms and normal-style LoRA for geometry and texture optimization, using visual comparison when excluding the component.

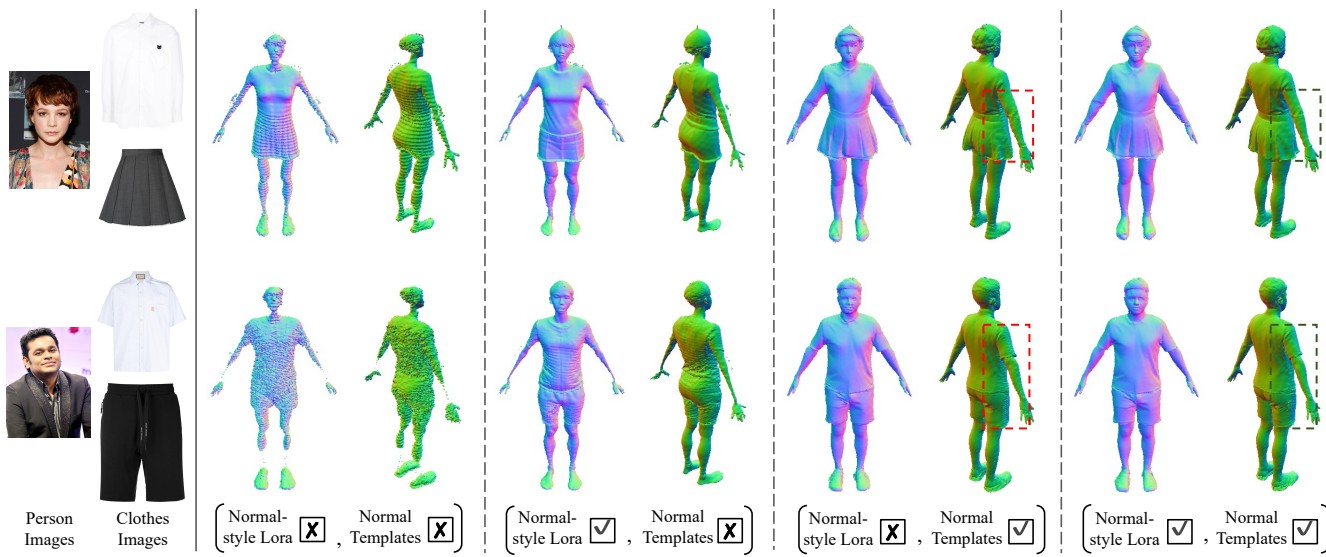

**Figure 6: Ablation study for geometry optimization.**

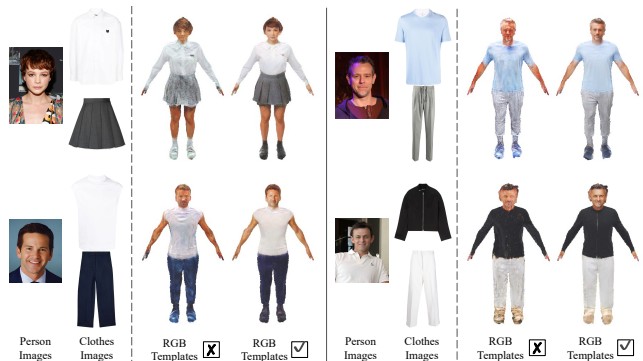

**Figure 7: Ablation study for texture optimization.**

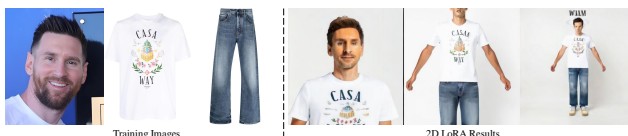

**Figure 8: Results of multi-concept LoRA. With complicated logos, LoRA fails to keep logo texture completely.**

For geometry optimization, as shown in Figure 6 , without using normal-style LoRA or normal templates as optimization constraints, the surface of the learned geometry is coarse with artifacts. Either adding the normal-style LoRA or using the normal templates in the geometry optimization procedure can smooth the geometry surface. By jointly using normal-style LoRA and normal templates, the geometry surface can be smoother while preserving the clothes characteristics in the input images.

For texture optimization, as shown in Figure 7, without using the RGB templates as optimization constraints, the learned texture is noisy (e.g., unclear face region) and fails to preserve the characteristic of inputs image (e.g., incorrect clothes color). By using the RGB templates during optimization, DreamVTON can generate 3D humans with high-quality texture and retain the input images' characteristics (i.e., person identity, clothes style).

## 5 LIMITATION

The texture detail of clothes in DreamVTON's result is derived from the generative priors from multi-concept LoRA. However, existing

LoRA fails to preserve the texture detail for complicated logos, (as shown in Figure 8), thus preventing DreamVTON from generating detailed textures that are completely consistent with input images. This problem could be alleviated by balancing the personalization and generalizability of personalized SD, which is widely explored in the diffusion-based models.

## 6 CONCLUSION

We propose a new method for 3D virtual try-on task, named DreamVTON, which is capable of generating the 3D human using person images, clothes images, and a text prompt as inputs. DreamVTON employs an SDS-based framework and disentangles the 3D try-on task as the geometry and texture separately. Specifically, DreamVTON introduces a personalized SD with multi-concept LoRA and Densepose-guided ControlNet to provide powerful pose consistent priors for 3D human optimization. DreamVTON designs a templated-based optimization mechanism for precise geometry and texture learning to avoid the degraded multi-view priors from personalized SD. In addition, DreamVTON integrates a normal-style LoRA into personalized SD during geometry optimization, which further facilitates smooth and accurate geometry. Extensive experiments demonstrate that our DreamVTON outperforms existing baselines in terms of geometry and texture model, and can customize high-quality 3D humans with diverse clothes, preserving the person's identity and clothes style.

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
