# OpenReview forum: "DreamVTON: Customizing 3D Virtual Try-on  with Personalized Diffusion Models"
_acmmm.org/ACMMM/2024/Conference — MM2024 Poster_

### Official Review · Reviewer_8xvm · 2024-05-23

**Rating:** 4
**Confidence:** 3

**Summary:**

This paper proposes an image-based 3D Virtual Try-On method, based on the DMTet geometry representation and SDS loss to perform two-stage 3D human generation. Specifically, the proposed method utilizes a Densepose-based ControlNet and multi-concept LoRA to generate pose-conditioned RGB/normal images with desired clothes and faces, which serve as the templates to optimize the 3D human. It also proposes a normal-style LoRA to enhance the geometry. The experiments validate the effectiveness of the proposed method.

**Strengths:**

1. Without the need for a large image-3D dataset for training, this method can be used to generate 3D virtual try-on results given several images through LoRA fine-tuning.
2. This paper is well-structured and easy to read.

**Limitations:**

1.  Almost all the 3D humans of the video demos provided in the supplementary material show inconsistent textures at the seams between the front and back views.
2.  Although this work is designed to solve 3D virtual try-on, it cannot specify the user's body shape with merely face images. I notice that the results demonstrated in the paper all have mean body shapes. It seems that using lora combined with Text-to-3D human reconstruction for virtual try-on is limited, as it performs worse in preserving garment texture and identity features. Experiments lack comparisons with other VTON methods to demonstrate its effectiveness on this task.

**Suitability:**

3

---

### Official Review · Reviewer_ZSgB · 2024-05-25

**Rating:** 5
**Confidence:** 3

**Summary:**

This paper proposes a 3D Virtual Try-On model that employs multiple personalized diffusion models to separately generate the geometry and texture of digital humans. The experimental results demonstrate the superior effectiveness of the DreamVTON model.

**Strengths:**

1. The final results significantly surpass the baseline model in terms of both geometry and texture. The alignment between geometry and texture is commendably achieved, showing a notable improvement over existing methods.
2. The adaptation of the 3D generation model for the virtual try-on domain is comprehensive and well-executed, making a decent contribution to the field.

**Limitations:**

1. The selection of human images for evaluation and demonstration shows an uneven distribution in terms of skin color and gender. Additionally, a greater variety of clothing choices could have been included to better demonstrate the robustness of the method.
2. There are problems of the deformation of hands in the generated results, and the clothing colors tend to mistakenly appear on the hands, which is more evident in the demo video.
3. There are still several questions that the authors did not mention: 1) How does this model compare in performance with traditional 3D virtual try-on solutions? 2) What is the time required to generate a single instance throughout the entire pipeline?
4. I notice that a similar article titled "AvatarFusion: Zero-shot Generation of Clothing-Decoupled 3D Avatars Using 2D Diffusion" was presented at MM2023 last year, which could be relevant for reference in further discussions of this topic.

**Suitability:**

3

---

### Official Review · Reviewer_VVHv · 2024-05-26

**Rating:** 4
**Confidence:** 3

**Summary:**

This paper introduces DreamVTON, a novel method for customizing 3D human try-on models. DreamVTON generates both the geometry and texture of a human figure wearing selected clothes.

**Strengths:**

1) The method is based on LoRA, making it lightweight.

2) The visual results are impressive.

3) In terms of quantitative metrics and user evaluations, DreamVTON outperforms the baselines.

**Limitations:**

1) The FID score is too high, indicating a need for more results to make the findings convincing. I suggest the authors calculate the FID at least five times with different samples and report the mean and standard deviation.

2) The CLIP-score for both TeCH and DreamVTON is the same, which is unusual. It is possible that the human pose itself introduces a high degree of CLIP-score similarity, making this metric less meaningful for this task. I would like to see more detailed research on this metric.

**Suitability:**

3

---

### Official Review · Reviewer_Mucp · 2024-05-26

**Rating:** 4
**Confidence:** 3

**Summary:**

- This paper introduces DreamVTON, a custom 3D human try-on model. The author utilizes multi-concept LoRA and Densepose-guided ControlNet for creating a personalized SD. Additionally, a template-based optimization mechanism is introduced to enhance geometry and shape details. Lastly, a normal-style LoRA is integrated to improve the normal map.
- DreamVTON seems to be an enhanced version of HumanNorm for personalized people and clothes.

**Strengths:**

- The paper is well-written and easy to follow.
- I appreciate the motivation experiment in Fig. 2, which indicates a potential issue when using LoRA diffusion in multi-view generation.
- The proposed DreamVTON seems to resolve this issue in Fig. 4, showing high-quality 3D try-on results and outperforming other 3D human generation methods.

**Limitations:**

- In Sec. 3.1, the authors claim that “the multi-concept LoRA is trained by jointly using person images and clothes images”, which is not sufficient for reproduction. More details of the multi-concept LoRA are recommended to add.

- The authors point out the multi-view generation issue when using LoRA diffusion. Is this issue mitigated by the proposed multi-concept LoRA? The solution and discussion are missing.

- Fig. 4 shows the sample results of multi-concept LoRA and Normal-style LoRA.  However, the normal map seems not aligned with the image (e.g., hair, skirt, and expression). Will the misalignment supervision aggravate the over-smoothing problem in a generation?

**Suitability:**

3

---

### Meta-Review · Area_Chair_Su82 · 2024-07-01

**Recommendation:** Accept (Poster)
**Confidence:** 4

**Metareview:**

All reviewers lean to accept the paper. Most concerns are cleared in the rebuttal. AC agrees with the reviewers and recommends acceptance.